# NEURAL CLUSTERING BY PREDICTING AND COPYING NOISE

## ABSTRACT

We propose a neural clustering model that jointly learns both latent features and how they cluster. Unlike similar methods our model does not require a predefined number of clusters. Using a supervised approach, we agglomerate latent features towards randomly sampled targets within the same space whilst progressively removing the targets until we are left with only targets which represent cluster centroids. To show the behavior of our model across different modalities we apply our model on both text and image data and achieve very competitive results on MNIST against methods that require a predefined number of clusters. We also provide results against baseline models for fashion-MNIST, the 20 newsgroups dataset, and a Twitter dataset we ourselves create.[1].

## 1   INTRODUCTION

Clustering is one of the fundamental problems of unsupervised learning. It involves the grouping of items into clusters such that items within the same cluster are more similar than items in different clusters. Crucially, the ability to do this often hinges upon learning latent features in the input data which can be used to differentiate items from each other in some feature space. Two key questions thus arise: *How do we decide upon cluster membership?* and *How do we learn good representations of data in feature space?*

Spurred initially by studies into the division of animals into taxa (Sokol & Sneath, 1963), cluster analysis matured as a field in the subsequent decades with the advent of various models. These included distribution-based models, such as Gaussian mixture models (Duda et al., 1973); density-based models, such as DBSCAN (Ester et al., 1996); centroid-based models, such as k-means.[2] and hierarchical models, including agglomerative  (Orloci, 1967) and divisive models (Gower, 1967).

While the cluster analysis community has focused on the unsupervised learning of *cluster membership*, the deep learning community has a long history of unsupervised *representation learning*, yielding models such as variational autoencoders (Kingma & Welling, 2013), generative adversarial networks (Goodfellow et al., 2014), and vector space word models (Mikolov et al., 2013).

In this paper, we propose using noise as targets for agglomerative clustering (or NATAC). As in Bojanowski & Joulin (2017) we begin by sampling points in features space called *noise targets* which we match with latent features. During training we progressively remove targets and thus agglomerate latent features around fewer and fewer target centroids using a simple heuristic. To tackle the instability of such training we augment our objective with an auxiliary loss which prevents the model from collapsing and helps it learn better representations. We explore the performance of our model across different modalities in Section 3.

Recently, there have been several attempts at jointly learning both cluster membership and good representations using end-to-end differentiable methods. Similarly to us, Yang et al. (2016a) use a policy to agglomerate points at each training step but they require a given number of clusters to stop agglomerating at. Law et al. (2017) propose a form of supervised neural clustering which can then be used to cluster new data containing different categories. Liao et al. (2016) propose jointly

---

[1]IDs of tweets we used can be found in:  https://github.com/neuralclusteringNAT/paper-resources/tree/master/tweet_clustering

[2]A good historical review of k-means can be found in Hans-Hermann (2008).

learning representations and clusters by using a k-means style objective. Xie et al. (2016) introduce deep embedding clustering (DEC) which learns a mapping from data space to a lower-dimensional feature space in which it iteratively optimizes a clustering objective (however, as opposed to the hard assignment we use, they optimize based on soft assignment).

Additionally, there have been unsupervised clustering methods using nonnegative low-rank approximations (Yang et al., 2016b; 2012) which perform competitively to current neural methods on datasets such as MNIST.

Unlike all of the above papers, our method does not require a predefined number of clusters.

## 2 MODEL

We begin by discussing the use of noise as targets (NAT) introduced in Bojanowski & Joulin (2017) which is crucial to the understanding of our model. We then describe the intuition behind our approach, then proceed to describe the mechanism itself.

### 2.1 NOISE AS TARGETS (NAT)

Bojanowski & Joulin (2017) proposed a new form of unsupervised learning called "Noise as Targets" (NAT), aimed at extracting useful features from a set of objects.[3] Roughly speaking, this approach selects a set of points, referred to as "targets" uniformly at random from the unit sphere. It then aims to find a mapping from the raw representations of objects to points on the unit sphere, so that these points are close to the corresponding targets; the correspondence (matching) between mapped representations and the target points is done so as to minimize the sum of the distances between the mapped points and the targets. The intuition behind the NAT approach is that the model learns to map raw inputs to the latent space in a way that both covers the entire latent space well, and that places "similar" inputs in neighborhoods of similar targets.

More formally, the NAT task is to learn an encoder function $f_\theta : X \rightarrow Z$ from input space to a latent representation space. The objective is to minimize the $L_2$ loss between representations $z_i \in Z$, where points in Z are unit normalized, and corresponding targets $y_k \in Y$ where the targets are uniformly sampled from the $L_2$ unit sphere (thus inhabiting the same space as the $z_i$).

Instead of being tied to corresponding representations as in classic regression, during the model fitting, inputs are considered in batches. Each batch consists of inputs and targets; when processing a batch the targets in the batch are permuted so as to minimize the batch-wise loss. To compute the optimal (loss-minimizing) one-to-one assignment of latent representations and targets of a batch, the Hungarian Method is used (Kuhn, 1955). This target re-assignment pushes the representations $z_i$ of similar inputs $x_i \in X$ to neighborhoods of similar targets $y_k$.

Importantly, every example must be paired to a single noise target. Minimizing the $L_2$ loss requires each latent representation to be close to its assigned target. Therefore, the model learns a mapping to latent space that very closely matches the distribution of the noise targets.

The motivation behind using NAT was to learn unsupervised features from input data. The authors show their method performs on par with state-of-the-art unsupervised representation learning methods.

### 2.2 NOISE AS TARGETS FOR AGGLOMERATIVE CLUSTERING (NATAC)

Viewed from a clustering perspective we can think of the targets $y_i$ as cluster centroids to which the latent representations $z_i$ are (one-to-one) assigned. Note that although the method introduced in Bojanowski & Joulin (2017) brings the representations of similar $x_i$ closer together (by matching and moving them closer to neighborhoods of similar targets) it cannot produce many-to-one matchings or match multiple similar $z_i$ with a single centroid thus forming a cluster. Simply changing the re-assignment policy to allow for many-to-one matchings is difficult because it causes the model to

---

[3]We include a brief technical description of NAT here to make this paper self-contained. The original paper Bojanowski & Joulin (2017) has a detailed description of the approach and the reasoning behind it.

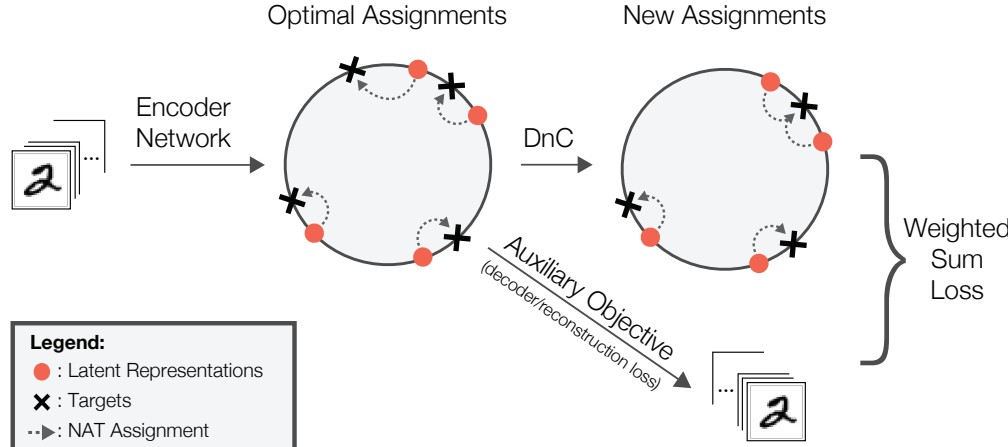

Figure 1: Training diagram for a NATAC model. In this case the model is clustering MNIST digits and uses an autoencoder loss as the auxiliary objective. The latent representation at the top of the circle is assigned to the top-left target, even though the target at the top-right of the circle is nearer. In this case, The delete-and-copy policy removes the target on the top-left of the sphere and copies the target at the top-right. This leads to an agglomeration of the two latent representations at the top of the sphere into the same cluster.

collapse in on a single target. In this paper we use the above to propose a new form of neural clustering where we progressively delete targets over time and re-assign representations to other nearby targets, all while keeping the model stable using an auxiliary objective.

**Delete-and-copy** To be able to cluster latent representations we need a way of assigning them to cluster centroids. We propose doing this via an additional delete-and-copy step which allows for many-to-one matchings. Similarly to the NAT method, we first assign representations $\mathbf{z}$ to targets $\mathbf{y}$ using the Hungarian method. In some cases, the optimally assigned target $y^{\text{opt}}$ is not the nearest target to a latent representation $z$. In this case, we remove the assigned target and reassign a *copy* of the nearest target for that $z$ with some probability $\alpha$ as in Algorithm 1 (see also Figure 1). This has the effect of not only reassigning targets so as to minimize the distance between matched pairs, but also to encourage the model to allow similar examples to be assigned to the same target. The new assignments are denoted as $y_i^{\text{new}}$. The loss is then defined as:

$$\mathcal{L}_{\text{NAT}}(\theta|x;y) = \sum_i \|y_i^{\text{new}} - z_i\|_2 \tag{1}$$

**Auxiliary objective** To prevent the model from collapsing to a single point, we introduce an auxiliary objective in addition to the loss between representations and targets. In our case we set the auxiliary objective $\mathcal{L}_{\text{aux}}$ to be the reconstruction loss $\sum_i \|x_i - f_{\text{dec}}(z_i)\|_2$ where $f_{\text{dec}}$ is some decoder network. The final objective $\mathcal{L}$ is then a weighted sum of the NAT loss $\mathcal{L}_{\text{NAT}}$ (which in our case is the $L_2$ loss) and the auxiliary objective $\mathcal{L}_{\text{aux}}$:

$$\mathcal{L}(\theta|x;y) = \lambda \mathcal{L}_{\text{NAT}}(\theta|x;y) + \mathcal{L}_{\text{aux}}(\theta|x) \tag{2}$$

Importantly, the auxiliary objective not only prevents the model from collapsing, it also informs how the model clusters. For example, when clustering images the reconstruction loss encourages the model to cluster on similar pixel values. Alternative forms of auxiliary objectives could allow for a discriminator loss or a classification loss for tackling semi-supervised settings. As our goal is unsupervised clustering, we only consider the reconstruction loss.

**Algorithm 1:** The delete-and-copy policy used in our experiments.

---

**Input** : A batch of latent representations $\mathbf{z}$, the optimal NAT assignment $\mathbf{y}^{\text{opt}}$, and a probability of copying $\alpha$

**Output:** A new NAT assignment $\mathbf{y}^{\text{new}} = \left( y_1^{\text{new}} \ldots y_i^{\text{new}} \ldots y_N^{\text{new}} \right)$

**for** $z_i$ *in* $z$ **do**

    $p :=$ sample from $U(0, 1)$

    $y^{\text{nearest}} :=$ nearest target to $z_i$ **if** $p < \alpha$ **then**

        ($y_i^{opt}$ *is then deleted, and* $y^{nearest}$ *is copied and assigned to* $z_i$)

        $y_i^{\text{new}} := y_i^{\text{nearest}}$

    **else**

        $y_i^{\text{new}} := y_i^{\text{opt}}$

    **end**

**end**

**return** $\left( y_1^{\text{new}} \ldots y_i^{\text{new}} \ldots y_N^{\text{new}} \right)$

---

**Model definition** During initialization each example $x_i$ in the dataset is paired with a random target $y_i$, uniformly sampled from a $d$-dimensional sphere. The NATAC model is then trained using mini-batches from the dataset. Each training step can be broken down as follows:

1. **Forward step:** The examples $\mathbf{x}$ from a random batch of example-target pairs $\left( (x_1, y_1) \ldots (x_i, y_i) \ldots (x_N, y_N) \right)$ are embedded using an encoder function $f_\theta : \mathbb{R}^n \rightarrow \mathbb{R}^d$ into corresponding latent representations $z_i \in \mathbb{R}^d$ where $\|z_i\| = 1$ for all $i$.

2. **Re-assignment step:** Using the Hungarian method from Kuhn (1955), the representations $\mathbf{z}$ are optimally one-to-one matched with targets $\mathbf{y}$ so as to minimize the total sum of distances between matched pairs in the batch. The newly assigned example-target pairing $\left( (x_1, y_1^{\text{opt}}) \ldots (x_i, y_i^{\text{opt}}) \ldots (x_N, y_N^{\text{opt}}) \right)$ has the permutation of labels within the batch to minimize the batch-wise loss.

3. **Delete-and-copy (DnC):** With a probability $\alpha$, delete the optimal target $y_i^{\text{opt}}$ and instead assign the nearest target to $z_i$ (which may be the same target) for each example-target pair in the batch producing $\left( (x_1, y_1^{\text{new}}) \ldots (x_i, y_i^{\text{new}}) \ldots (x_N, y_N^{\text{new}}) \right)$ as described in Algorithm 1 (see also Figure 1).

4. **Train and update step:** The $L_2$ loss between targets and latent representations is taken and combined with the auxiliary loss. Gradients w.r.t. $\theta$ are then taken and back-propagated along. Notice that although the (re)-assignment step follows a non-differentiable policy, the model is still end-to-end differentiable. Finally, the new example-target assignments are kept after the training step, and persist into the next training step where they are reassigned again.

## 2.3 IMPLEMENTATION DETAILS

**Stopping criterion** During training the number of unique targets is tracked. We stop training when the number of unique targets stops decreasing after an epoch of training.

**Multi-stage training** We found that an initial period of training where the auxiliary objective is prioritized (i.e. the NAT loss is multiplied by a very small coefficient) and the DnC policy is not used, improved overall performance. Transitioning to a higher NAT loss and turning on the delete-and-copy policy later on in training increased the stability of our model. We therefore propose training NATAC models as follows:

1. **Warm-Up stage**: $\alpha = 0$ and $\lambda$ is very small.

2. **Transition stage**: $\alpha$ increases gradually from 0 to 1, $\lambda$ also increases gradually to a larger value (approximately $100\times$ larger than its initial value).

3. **Clustering stage**: $\alpha = 1$, $\lambda$ is large.

| | MNIST | | | | Fashion-MNIST | | | |
|---|---|---|---|---|---|---|---|---|
| $d$ | Centroids | NATAC | NATAC-k | AE-k | Centroids | NATAC | NATAC-k | AE-k |
| 3 | 2251 | 0.455 | 0.455 | 0.452 | 2330 | 0.442 | 0.441 | 0.439 |
| 8 | 120 | 0.820 | 0.681 | 0.650 | 101 | 0.614 | 0.542 | 0.538 |
| 9 | 308 | 0.782 | 0.640 | 0.611 | 77 | **0.638** | 0.562 | 0.557 |
| 10 | 152 | **0.878** | 0.704 | 0.649 | 44 | 0.627 | 0.570 | 0.564 |
| 11 | 123 | 0.865 | 0.702 | 0.696 | 45 | 0.619 | 0.566 | 0.580 |
| 12 | 56 | 0.767 | 0.688 | 0.683 | 20 | 0.607 | 0.572 | 0.609 |

Table 1: NMI scores from varying the dimensionality of the latent space in the NATAC and baseline models. The baselines use k-means with the same number of clusters as the repsective NATAC model converged to. We include NATAC models with a latent dimensionality of $d = 3$, whose latent representations can be viewed without dimensionality reduction. Appendix A contains links to the visualizations hosted on the TensorFlow embedding projector.

**Dimensionality of the latent space** In all of our experiments, we found that the best performing models tend to have a latent space dimensionality between $4$ and $12$.

At dimensionalities much larger than this, the model collapses to very few points during the transition stage of training, possibly due to the high expressiveness of the latent space. On the other hand, using a low dimensional representation results in an information bottleneck too small to sufficiently learn the auxiliary objective. For example, when clustering tweets from our Twitter dataset, a latent space of two dimensions was too small for the decoder to reliably reconstruct tweets from latent vectors. With an auxiliary objective that cannot be effectively learned, centroids collapse to a single point.

## 3 EXPERIMENTS

We now describe the datasets and evaluation metrics used in our experiments followed by the presentation and analysis of our results in comparison to others. The full details regarding the hyperparameters used in our experiments can be found in appendix D.

**Datasets** We evaluate our models on four different datasets - two image and two text datasets. For images we use MNIST (LeCun et al., 1998) and Fashion-MNIST (Xiao et al., 2017). For text we use 20 Newsgroups (Joachims, 1996) and a Twitter dataset which we gather ourselves.

### 3.1 EVALUATION

Our key evaluation metric is the normalized mutual information (NMI) score (Strehl & Ghosh, 2002) which measures the information gained from knowing cluster membership whilst taking the number of clusters into account. Values of NMI range from $0$, where clustering gives no information, to $1$, where clustering gives perfect information i.e. cluster membership is identical to class membership.

In our experiments, we train models on the concatenation of the train and validation sets and evaluate them on the test set. This is done by computing the latent representations of test examples and then assigning them to their nearest respective centroids, then computing the NMI score. Additionally, we provide classification error scores on the MNIST dataset to compare ourselves to other related methods. We also compare our model to clustering methods trained on the 20 Newsgroups dataset. The guiding motivation behind our experiments is to analyze how well our models learns cluster membership *and* latent representations.

### 3.2 MNIST

Introduced by LeCun et al. (1998), MNIST is canonical dataset for numerous machine learning tasks including clustering. MNIST has ten classes, corresponding to the ten digits 0-9, and contains 60,000 train and 10,000 test examples.

We train our model with an auxiliary reconstruction loss and use small convolutional architectures (see Figure 5) for our encoder and decoder networks. As points of comparison we also provide results of using k-means on the latent representations learned by our model (NATAC-k in Table 1) and k-means on representations learned by a simple autoencoder with the same encoder and decoder architecture (AE-k in Table 1).

Table 1 shows our model's performance is best when $d = 10$ and worse for much lower or higher values of $d$. This indicates that the dimensionality of the latent space impacts our model's performance (see Section 2.3 for further discussion).

The ability of our model to cluster MNIST examples well is shown by two keys results. First, it beats both NATAC-k and AE-k (Table 1). Second, it achieves very competitive results when compared to other methods (see NMI column in Table 2). To re-iterate, other clustering techniques cited in Table 2 required a predefined number of clusters (in the case of MNIST k=10).

We note that NATAC-k beats AE-k indicating that our model learns representations that suit k-means clustering more than a simple autoencoder. However, we note that this is not consistent across all modalities in this paper (see results in section 3.4).

Finally, we discuss the number of centroids our model converges on (see Table 1). We show that our model is successfully capable of finding centroids that represent different digits, as shown in the top row of Figure 2. However, the model also learns centroids that contain very few examples for which the decoded images do not represent any handwritten digits, as shown in the second row of Figure 2. Even with these "dead centroids", the model still performs well. Indeed, the twelve most dense centroids contain $98\%$ of all of the examples in MNIST (out of a total of 61).Interestingly, the model also differentiates between ones with different slopes. This suggests that the latent representations of these digits are sufficiently far apart to warrant splitting them into different clusters.

### 3.3 FASHION-MNIST

Introduced in Xiao et al. (2017), Fashion-MNIST is a convenient swap-in dataset for MNIST. Instead of digits the dataset consists of ten different types of clothes. There are 60,000 train and 10,000 test examples just like in MNIST. Fashion-MNIST is generally considered more difficult than MNIST, with classifiers scoring consistently lower on it.[4]

The model and analysis from the previous section carry over for fashion-MNIST with a few additional important points. First, the differences between NATAC-k and AE-k are less pronounced (see Table 1) in fashion-MNIST which indicates that the representations learned by NATAC in comparison to a simple autoencoder are not as important for k-means clustering. Interestingly, our model still outperforms both NATAC-k and AE-k, with one exception being when $d = 12$.

Qualitatively, Figure 2 shows that the model separates garments into slightly different categories than the labels in the dataset. For example, the most dense cluster seems to be a merging of both "pullovers" and "shirts", suggesting that the model finds it difficult to separate the two different garments. We ourselves find it difficult to discriminate between the two categories, as the low-resolution images do not easily show whether or not the garment has buttons. Additionally, the "sandal" class has been split into two separate clusters: flip-flops and high-heeled shoes with straps. This indicates that our model has found an important distinction between these two type of shoes, that the original Fashion-MNIST labels ignore. Similarly to MNIST, our model also learns "dead clusters", which the model does not decode into any discernible garment. Further visualizations of these experiments can be found appendix section A.

### 3.4 LARGE DOCUMENT CLUSTERING ON 20 NEWSGROUPS

Introduced in Joachims (1996) the 20 Newsgroups dataset is a collection of 18,846 documents pertaining to twenty different news categories. We use the commonly used 60:40 temporal train-test split. Interestingly, because of the temporal split in the data, the test set contains documents which differ considerably from the train set. We calculate NMI on the news categories in the dataset.

---

[4]A MNIST vs Fashion-MNIST comparison: https://github.com/zalandoresearch/fashion-mnist#benchmark

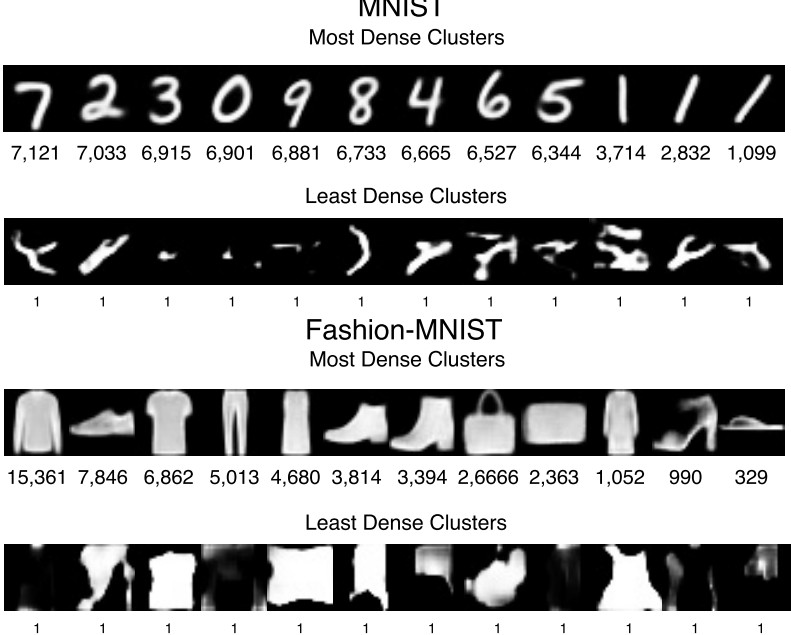

Figure 2: Outputs of the decoder when fed centroids from the MNIST experiment (top) and Fashion-MNIST experiment (bottom). The models used are those with the highest NMI, $d = 10$ and $d = 9$ for MNIST and Fashion-MNIST respectively. The top rows show decoded images from the centroids of the densest clusters (cluster sizes are shown underneath). Similarly, the bottom rows show decoded images from the least dense clusters.

| Clustering Algorithm | NMI | Classification Error (%) |
|---|---|---|
| k-means (MacQueen et al., 1967) On Raw Pixels | 0.500 | 41.50 |
| DEC (Xie et al., 2016) | - | 15.70 |
| Adversarial Autoencoders (30 clusters) (Makhzani et al., 2015) | - | 4.10 |
| Deep Gaussian Mixture VAE (Dilokthanakul et al., 2016) | - | 3.08 |
| IMSAT Hu et al. (2017) | - | **1.6** |
| Autencoder based clustering (Song et al., 2013) | 0.669 | - |
| Task-specific Clustering With Deep Model (Wang et al., 2016) | 0.651 | - |
| Agglomerative Clustering Using Average Linkage (Jain et al., 1999) | 0.686 | - |
| Large-Scale Spectral Clustering (Chen & Cai, 2011a) | 0.706 | - |
| Yang et al. (2016a) | 0.906 | - |
| Yang et al. (2016a) With Re-clustering | 0.913 | - |
| DCD Yang et al. (2016b) | **0.93** | 3 |
| NATAC Autoencoder (Converged to 61 clusters) | 0.915 | 2.78 |

Table 2: Comparison of our best performing NATAC model (with $d = 10$) on the entire MNIST dataset. NMI and classification error are calculated from the entire data set. We report the evaluation metric used by the authors of each respective model. Precision of values are the same as those reported by the original paper. Note that many of the best-performing methods (DCD, IMSAT, Adversarial Autoencoders) also assume a uniform class distribution along with a pre-set number of clusters.

We use an auxiliary reconstruction loss and a two layer fully connected network for both the encoder and decoder, both with hidden layer sizes of 256 and ReLU nonlinearities. We represent each article as an L2 normalized term-frequency-inverse-document-frequency (TF-IDF) vector of the 5000 most occurring words in the train set.

| | **Spherical TF-IDF Autoencoder** | | | | **Sph. k-means** | |
|---|---|---|---|---|---|---|
| $d$ | Centroids | NATAC | NATAC-k | AE-k | Clusters | NMI |
| 2 | 218 | 0.271 | 0.162 | 0.219 | 20 | 0.216 |
| 3 | 416 | 0.350 | 0.220 | 0.320 | 50 | 0.178 |
| 4 | 500 | **0.413** | **0.283** | 0.359 | 100 | 0.306 |
| 5 | 371 | 0.406 | 0.270 | 0.365 | 500 | 0.280 |
| 6 | 321 | 0.378 | 0.255 | **0.379** | 1000 | **0.337** |

Table 3: Clustering results for the 20 Newsgroups dataset.

| Method | NMI | Method | NMI |
|---|---|---|---|
| 1-Spec (Hein & Bühler, 2010) | 0.08 | LSD (Arora et al., 2011) | 0.44 |
| MTV (Bresson et al., 2013) | 0.13 | PLSI (Hofmann, 1999) | 0.47 |
| SSC (Elhamifar & Vidal, 2009) | 0.29 | LSC (Chen & Cai, 2011b) | 0.48 |
| PNMF (Yang & Oja, 2010) | 0.37 | Ncut (Shi & Malik, 2000) | 0.52 |
| ONMF (Ding et al., 2006) | 0.38 | NSC (Ding et al., 2008) | 0.52 |
| k-means (MacQueen et al., 1967) | 0.44 | DCD (Yang et al., 2016b) | **0.54** |
| NATAC Autoencoder (425 clusters) | 0.479 | | |

Table 4: Comparison of our best performing NATAC model (with $d = 4$) on the entire 20 Newsgroups dataset. NMI is calculated from the entire data set. Figures for other methods taken from Yang et al. (2016b)

Along with NATAC-k and AE-k comparisons, we also use a spherical k-means model. Spherical k-means is a commonly used technique of unsupervised document clustering, a good description of it can be found in Buchta et al. (2012).

Table 3 shows how the performance of the each model varies with different dimensionalities of the latent space. The best NATAC models with a latent dimensionality of 3 to 6 centroids outperform a spherical k-means model with 1000 clusters, far more clusters than any of the NATAC models.

Although we could not find any neural clustering techniques which report performance on 20 Newsgroups, many non-neural methods report NMI on the whole dataset. Table 4 shows the NATAC model performs competitively to these methods. However, our method does converge on a higher number of clusters (other methods are trained with a pre-defined number of 20 clusters).

## 3.5 CLUSTERING TWEETS

To further explore the performance of our model on text data we build a dataset of 38,309 ASCII-only tweets of English speaking users containing exactly one hashtag. The dataset has 647 different hashtags, with each hashtag having at least ten tweets containing it. We use 10,000 of the tweets as the test set. As a preprocessing step, URLs and hashtags are replaced with special characters. We calculate NMI on the hashtag used in each tweet.

| | **Character-based RNN** | | | | **Sph. k-means** | |
|---|---|---|---|---|---|---|
| $d$ | Centroids | NATAC | NATAC-k | AE-k | Num. Clusters | NMI |
| 7 | 1,574 | 0.646 | 0.671 | **0.663** | 700 | 0.595 |
| 8 | 1,314 | **0.656** | **0.675** | 0.650 | 1,000 | 0.615 |
| 9 | 1,020 | 0.605 | 0.657 | 0.644 | 1,500 | 0.628 |
| 10 | 269 | 0.456 | 0.522 | 0.517 | 2,000 | **0.649** |

Table 5: Clustering results for the Twitter dataset. Spherical k-means models trained with a vocabulary size of 5,000 (same as the 20 Newsgroups baselines).

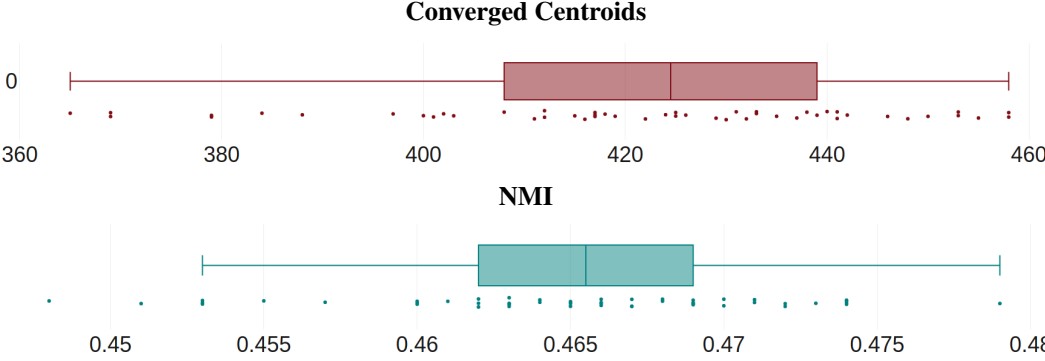

Figure 3: Box plots the converged number of clusters (above) and NMI (below) of running 50 training runs on the whole of 20 Newsgroups. Every run has the same hyperparameters as the best performing NATAC model from the previous experiments ($d = 4$).

We train a character-based Sequence-to-Sequence autoencoder on the Twitter dataset. Just as before we use an auxiliary reconstruction loss. We set the encoder to be a bidirectional GRU with a hidden size of 64 followed by a fully connected layer which takes the GRU's final output and maps it to the latent space. The decoder uses a fully connected layer to map the latent vectors to a 128 dimensional vector which is then used as the initial hidden state for the decoder GRU.

Similarly to section 3.4, we compare our approach to using spherical k-means along with the NATAC-k and AE-k baselines. As shown in table 3.5, we see that NATAC-k outperforms NATAC and AE-k models on all of the reported latent dimensionalities. This suggests that the latent mapping learned by the NATAC models does improve on a vanilla autoencoder, but the centroid assignment of a trained NATAC model is less effective than using k-means. Finally, all of the neural models outperform the spherical k-means baseline. However, this baseline is much more competitive to the neural methods reported in this experiment than those reported in the 20 Newsgroups experiments.

## 4 ROBUSTNESS EXPERIMENTS

In this section, we explore how sensitive the NATAC framework is to changes in our hyperparameters. For these experiments, we use the model and dataset from the 20 Newsgroups experiments. We take the best performing hyperparameters from these experiments ($d = 4$, see Table 3) and observe how the end performance of the model is affected by changing the hyperparameters. We show that our method is reasonably robust to differing values of hyperparameters, although extreme changes, such as skipping pre-training, do adversely affect performance.

### 4.1 VARIABILITY IN TRAINING

The NATAC training method contains four sources of randomness: (1) parameter initialization (2) the delete-and-copy policy (3) random batching (4) the sampling of noise as targets. We train 50 NATAC models with the same hyperparameters (but different random seeds) and measure the variation in NMI and the number of converged clusters. Figure 3 shows the variability of NMI and the converged number of clusters from training the best performing model on the 20 Newsgroups dataset. We observe that the NMI varies with a small standard deviation of 0.007 (mean 0.465) regardless of how many clusters the model converged to. In contrast, we observe a higher relative standard deviation of 24 with the converged number of clusters (mean 420). Qualitatively, we observe that the variance in the number of converged clusters is mostly due to dead centroids.

### 4.2 VARYING THE LENGTH OF PRE-TRAINING

The value of $\alpha$ varies throughout the training of a NATAC model. In our experiments, we initially set $\alpha$ to zero for a period of pre-training, after which we incrementally increase the value to 1 over several epochs. Table 6 shows that the amount of pre-training *does* impact the end NMI of the model,

| Epochs of Warm-up Training | NMI | Number of Centroids |
|---|---|---|
| 0 | 0.311 | 15 |
| 10 | 0.351 | 84 |
| 100 | 0.397 | 298 |
| 1,000 | 0.401 | 481 |
| 10,000 | 0.398 | 555 |

Table 6: Mean NMI and converged number of centroids when training a NATAC model with varying amounts of pre-training. Mean taken from 5 consecutive runs using the same hyperparameters. Models trained on the train set of 20 Newsgroups and evaluated on the test set.

| $\lambda_{\text{final}} \setminus \lambda_{\text{inital}}$ | $10^{-5}$ | $10^{-4}$ | $10^{-3}$ |
|---|---|---|---|
| $10^{-1}$ | 0.402 | 0.390 | 0.432 |
| $10^{-2}$ | 0.397 | 0.393 | 0.427 |
| $10^{-3}$ | 0.396 | 0.394 | 0.431 |

| $\lambda_{\text{final}} \setminus \lambda_{\text{inital}}$ | $10^{-5}$ | $10^{-4}$ | $10^{-3}$ |
|---|---|---|---|
| $10^{-1}$ | 396 | 477 | 862 |
| $10^{-2}$ | 471 | 497 | 837 |
| $10^{-3}$ | 444 | 506 | 773 |

Table 7: Mean NMI (left) and converged number of clusters (right) of NATAC models with different values for $\lambda_{\text{inital}}$ and $\lambda_{\text{final}}$. Mean values taken from 5 consecutive runs using the same hyperparameters. Models trained on the train set of 20 Newsgroups and evaluated on the test set.

but after 100 epochs of pre-training, the model does not significantly benefit from any more pre-training. Interestingly, the longer the period of pre-training, the more clusters the model converges to. We believe that models which have longer to pre-train before clustering learn a more uniform mapping to latent space. When the clustering phase of training occurs, the latent representations are more uniformly spread across latent space, and thus agglomerate less readily.

## 4.3  Varying The Loss Coefficient $\lambda$

Alongside changing the value of $\alpha$, the coefficient for the NAT loss in NATAC models is also varied. Similarly to $\alpha$, we set $\lambda$ to a small value for warm-up stage of training, and then progressively increase $\lambda$ to a larger value afterwards. In the other experiments involving 20 Newsgroups, warm-up training uses $\lambda_{\text{inital}} = 10^{-4}$ and a final value of $\lambda_{\text{final}} = 10^{-2}$. The transition happens at the same time as the change in $\alpha$ during training.

Table 7 shows how the final NMI of NATAC models vary with differing values for $\lambda_{\text{inital}}$ and $\lambda_{\text{final}}$. We notice that the value of $\lambda_{\text{final}}$ does not seem to greatly impact the number of clusters or NMI, and that the smaller values of $\lambda_{\text{inital}}$ have very similar NMI scores and number of clusters. Interestingly, when trained with large $\lambda_{\text{inital}}$ of $10^{-3}$, the model scores higher in NMI than models with smaller $\lambda_{\text{inital}}$ and also converges to more clusters. Here, we believe that a larger $\lambda_{\text{inital}}$ forces the model to learn a more uniform mapping to latent space (to minimize the NAT loss) during the warm-up stage of training. Similar to increasing the length of pre-training, this causes the model to agglomerate less readily.

## 5  Conclusion

In this paper, we present a novel neural clustering method which does not depend on a predefined number of clusters. Our empirical evaluation shows that our model works well across modalities. We show that NATAC has competitive performance to other methods which require a pre-defined number of clusters. Further, it outperforms powerful baselines on Fashion-MNIST and text datasets (20 Newsgroups and a Twitter hashtag dataset). However, NATAC does require some hyperparameters to be tuned, namely the dimensionality of the latent space, the length of warm-up training and the values for the loss coefficient $\lambda$. However, our experiments indicate that NATAC models are fairly robust to hyperparameter changes.

**Future work** Several avenues of investigation could flow from this work. Firstly, the effectiveness of this method in a semi-supervised setting could be explored using a joint reconstruction and classi-

fication auxiliary objective. Another interesting avenue to explore would be different agglomerative policies other than delete-and-copy. Different geometries of the latent space could also be considered other than a unit normalized hypersphere. To remove the need of setting hyperparameters by hand, work into automatically controlling the coefficients (e.g. using proportional control) could be studied. Finally, it would be interesting to see whether clustering jointly across different feature spaces would help with learning better representations.

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

# Appendix

## A  VISUALIZATIONS OF MNIST/FASHION-MNIST LATENT REPRESENTATIONS

We have published the visualizations of MNIST/Fashion-MNIST representations on the TensorFlow embedding projector. We recommend selecting the "Spherize data" option, as this gives a much clearer view of the embeddings. The images used are taken from the test set of MNIST and Fashion-MNIST respectively. We include the NATAC models with the highest NMI from each dataset, along with the models with $d = 3$, which require no dimensionality reduction to visualize.

- MNIST $d = 3$:
  http://projector.tensorflow.org/?config=https://raw.githubusercontent.com/
  neuralclusteringNAT/paper-resources/master/mnist_embeddings/digits3/config.json

- MNIST $d = 10$:
  http://projector.tensorflow.org/?config=https://raw.githubusercontent.com/
  neuralclusteringNAT/paper-resources/master/mnist_embeddings/digits10/config.json

- Fashion-MNIST $d = 3$:
  http://projector.tensorflow.org/?config=https://raw.githubusercontent.com/
  neuralclusteringNAT/paper-resources/master/mnist_embeddings/fashion3/config.json

- Fashion-MNIST $d = 9$:
  http://projector.tensorflow.org/?config=https://raw.githubusercontent.com/
  neuralclusteringNAT/paper-resources/master/mnist_embeddings/fashion9/config.json

## B   EXAMPLES FROM THE FASHION-MNIST DATASET.

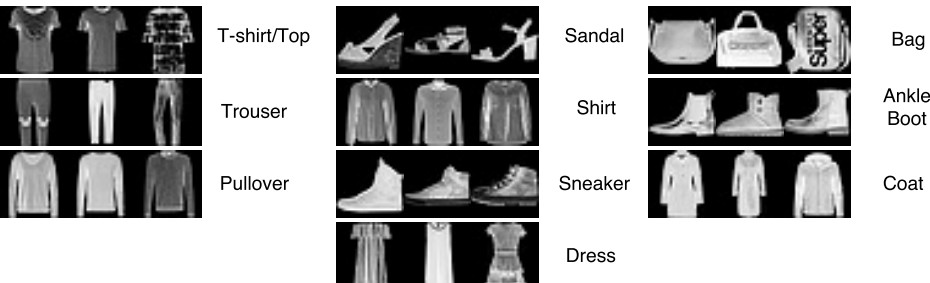

Figure 4:

## C   ADDITIONAL DISCUSSION ON TRAINING NATAC MODELS

### C.1   GEOMETRY OF LATENT SPACE

We experimented with using polar coordinates early on in our experiments. Rather than using euclidean coordinates as the latent representation, $z$ is considered a list of angles $\theta_1, \theta_2 \cdots \theta_n$ where $\theta_1 \cdots \theta_{n-1} \in [0, \pi]$ and $\theta_n \in [0, 2\pi]$. However, we found that the models using polar geometry performed significantly worse than those with euclidean geometry.

Additionally, we also experimented with not L2 normalizing the output of the encoder network. We hypothesized that the model would learn a better representation of the latent space by also "learning" the geometry of the noise targets. Unfortunately, the unnormalized representation caused the noise targets to quickly collapse to a single point.

## D   HYPERPARAMETERS FOR EXPERIMENTS

Although each different modality (monochrome images, bag-of-words, sequence of characters) uses a different set of hyperparameters, we follow a similar recipe for determining the values for each one:

- We use a large batch size of 100. This is so each batch has a representative sample of the targets to reassign in each training step.
- The warm-up period is calculated by observing when the auxiliary objective starts to converge.
- The final value for $\lambda$ in training is set so that the NAT loss was approximately 1% of the total loss.
- The initial value for $\lambda$ is set as approximately 1% of the final value of $\lambda$.
- The transition phase typically lasts 100 epochs of training.
- During the transition phase, the value of $\alpha$ is incrementally increased from 0 to 1.

We now explicitly list the hyperparameters used for each experiment:

### D.1   MNIST AND FASHION MNIST

- A batch size of 100.
- A warm-up period of $10 \times d$ epochs, during which $\lambda = 0.001$.
- A transition period lasts for 250 epochs, where $\lambda$ is incrementally increased to 0.25, and $\alpha$ is incremented from 0 to 1.
- The ADAM optimizer (Kingma & Ba, 2014) with a learning rate ($\alpha = 10^{-4}$)

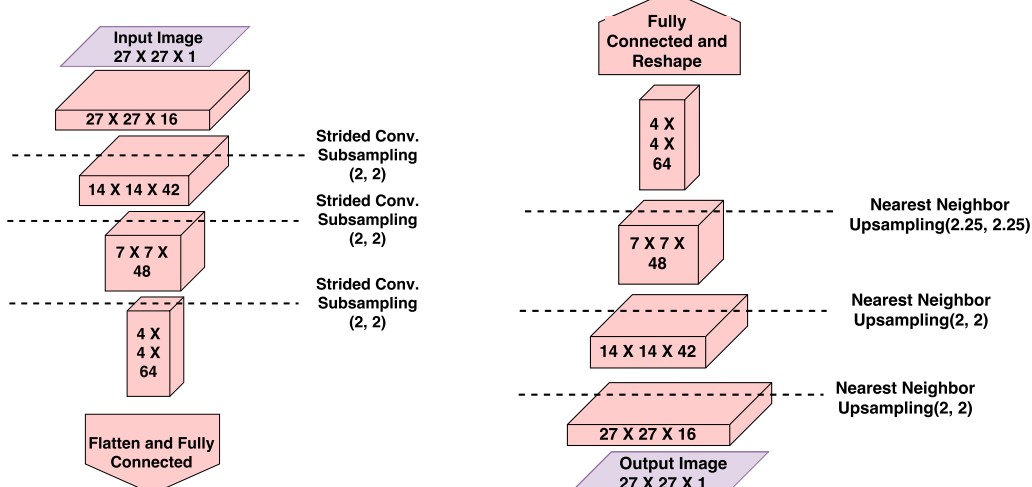

Figure 5: Architecture of the encoder (left) and decoder (right) used for the MNIST experiments. Between each subsampling layer in the encoder, a single convolution layer is applied with a filter shape of $3 \times 3$ with border padding to keep the same shape before and after the convolution. Similarly, in the decoder one transpose convolutional layer is applied between each upsampling layer, $3 \times 3$ filter shape and shape-preserving padding.

## D.2   20 NEWSGROUPS

- A batch size of 100.
- A warm-up period of $1,000$ epochs, during which $\lambda = 10^{-4}$.
- A transition period lasts for 100 epochs, where $\lambda$ is incrementally increased to $0.01$, and $\alpha$ is incremented from 0 to 1.
- The ADAM optimizer (Kingma & Ba, 2014) with a learning rate ($\alpha = 10^{-5}$).
- Dropout with a keep-probability of 0.95 in the hidden layers of the encoder and decoder.

## D.3   TWITTER DATASET

- A batch size of 100.
- A warm-up period of 100 epochs, during which $\lambda = 0.01$.
- A transition period lasts for 100 epochs, where $\lambda$ is incrementally increased to 1, and $\alpha$ is incremented from 0 to 1.
- The ADAM optimizer (Kingma & Ba, 2014) with a learning rate ($\alpha = 10^{-3}$).

