# OpenReview forum: "Neural Clustering By Predicting And Copying Noise"
_ICLR.cc/2018/Conference — Reject_

### Official Review · AnonReviewer3 · 2017-11-23
**overall algorithm is somewhat heuristic**

**Rating:** 5
**Confidence:** 4

**Review:**

This paper presents an algorithm for clustering using DNNs. The algorithm essentially alternates over two steps: a step that trains the DNN to predict random targets, and another step that reassigns the targets based on the overall matching with the DNN outputs. The second step also shrinks the number of targets over time to achieve clustering. Intuitively, the randomness in target may achieve certain regularization effect.

My concerns:
1. There is no analysis on what the regularization effect is. What advantage does the proposed algorithm offer to an user that a more deterministic algorithm cannot?
2. The delete-and-copy step also introduces randomness, and since the algorithm removes targets over time, it is not clear if the algorithm consistently optimizes one objective throughout. Without a consistent objective function, the algorithm seems somewhat heuristic.
3. Due to the randomness from multiple operations, the experiments need to be run multiple times, and see if the output clustering is sensitive to it. If it turns out the algorithm is quite robust to the randomness, it is then an interesting question why this is the case.
4. Does the  Hungarian algorithm used for matching scales to much larger datasets?
5. While the algorithm empirically improve over k-means, I believe at this point combinations of DNN with classical clustering algorithms already exist and comparisons with such stronger baselines are missing. The authors have listed a few related algorithms in the last paragraph on page 1. I think the following one is also relevant:
-- Law et al. Deep spectral clustering learning. ICML 2015.

---

> ### Author Response · Authors · 2017-12-08
> **Response**
>
> Thank you for the insightful comments!
>
> Regarding the points you have made:
>
> 1&2: In NATAC,  the targets are uniformly, randomly sampled from a unit-sphere (one for each example in the dataset). With the large size of the datasets in our experiments (between 20 and 70 thousand) the randomly sampled targets should very closely approximate a uniform distribution on the sphere. In the warm-up stage of training, we do not utilize the delete-and-copy mechanism, meaning that the initial objective is to both autoencode examples _and_ uniformly map the latent representations onto a unit sphere. Therefore, the latent representations serve as a lossy compression of the input data whilst incentivised to be spread uniformly over a unit sphere.
>
> Although the one of the objectives in the warm-up stage of training is to have a uniformly distributed latent representations, there will always be inconsistencies: As the reconstruction loss ‘encourages’ similar examples to have similar latent representations, the distribution of the latent representations will be denser in some regions and more sparse in others.
>
> In the transition and clustering stages of training, we then gradually perturb the distribution of the targets to more closely match the imperfect distribution of the latent representations (we use the heuristic delete-and-copy mechanism), and also allow for targets to agglomerate.
>
> The randomness comes from two sources:
> One being the initial random assignment of examples in the dataset X to the latent targets Y and training in mini-batches. As mentioned before, the warm-up stage of training is responsible for finding a good assignment of the targets to the input examples, and the targets are a very close approximation of a uniform distribution of points on a sphere.
>
> The delete-and-copy function can be seen as a way of gradually re-aligning the distribution of the targets more closely to the distribution of the latent representations made by the encoder. The randomness comes from the fact that, instead of deterministically, we randomly choose which targets to delete in training. However, the delete-and-copy mechanism is only stochastic during the transition-phase of training - after which we delete-and-copy with a probability of 1 (alpha = 1).
>
> Indeed, our delete-and-copy method is a randomized algorithm. Intuitively, what it tries to achieve is gradually remove targets from the less dense areas (and clone targets in the denser areas; in turn, this allows the latent representation more freedom (by making the constraints easier to meet), so it is easier to reconstruct the example from its latent representation.
>
> The random effect of shifting targets may help avoid overfitting (memorizing certain locations in latent space ‘off-by-heart’). In a way it is reminiscent to VAEs, where instead the latent representation is perturbed before it is passed to the decoder. We will include a brief discussion about this intuition in the paper - thank you for raising this!
>
> We do not know whether it is possible to achieve similar results with a deterministic perturbation algorithm. However, our delete-and-copy method has outperformed any of the deterministic heuristics we have tried (e.g. simple rules like removing one target from the least populated area and cloning one target in the most dense area). It is a very interesting question for future research to see whether more elaborate heuristics, and in particular deterministic ones, can yield results better than we’ve obtained using our simple delete-and-copy randomized rule.
>
> We are not aware other neural methods that do not require a set number of clusters (with the one exception being another paper submitted to this ICLR), so we were unable to comment on the difference between this method and more-deterministic methods.
>
>
> 3. We found the outcome of training a model with similar hyperparameters to be fairly similar in outcome. We plan on showing the variability of training the best performing model on 20 Newsgroups (as these models are quick to train) to empirically show this in the paper.
>
> 4: The hungarian method itself runs in O(N^3) complexity - significantly more efficient than a brute-force search (O(n!)). This would mean computing the optimal assignments over the whole dataset would be expensive. However, we train using mini-batches, meaning that the hungarian algorithm is only computed on a batch of data, not the whole dataset. This means that the runtime of a single forward/backward pass of a model does not change wrt the size of the dataset, as the batch size remains constant.
>
> 5: Thank you for bringing this paper to your attention, we will certainly mention this in our revision. Unfortunately, the paper does not report NMI on the datasets we do, so we are unable to compare performance with our method.

---

### Official Review · AnonReviewer2 · 2017-11-24
**interesting method while less satisfactory results**

**Rating:** 5
**Confidence:** 3

**Review:**

This ms presents a new clustering method which combines deep autoencoder and a recent unsupervised representation learning approach (NAT; Bojanowski and Joujin 2017). The proposed method can jointly learn latent features and the cluster assignments. Then the method is tested in several image and text data sets.

I have the following concerns:

1) The paper is not self-contained. The review of NAT is too brief and makes it too hard to understand the remaining of the paper. Because NAT is a fundamental starting point of the work, it will be nice to elaborate the NAT method to be more understandable.

2) Predicting the noise has no guarantee that the data items are better clustered in the latent space. Especially, projecting the data points to a uniform sphere can badly blur the cluster boundaries.

3) How should we set the parameter lambda? Is it data dependent?

4) The experimental results are a bit less satisfactory:
a) It is known that unsupervised clustering methods can achieve 0.97 accuracy for MNIST. See for example [Ref1, Ref2, Ref3].
b) Figure 3 is not satisfactory. Actually t-SNE on raw MNIST pixels is not bad at all. See https://sites.google.com/site/neighborembedding/mnist
c) For 20 Newsgroups dataset, NATAC achieves 0.384 NMI. By contrast, the DCD method in [Ref3] can achieve 0.54.

5) It is not clear how to set the number of clusters. More explanations are appreciated.

[Ref1] Zhirong Yang, Tele Hao, Onur Dikmen, Xi Chen, Erkki Oja. Clustering by Nonnegative Matrix Factorization Using Graph Random Walk. In NIPS 2012.
[Ref2] Xavier Bresson, Thomas Laurent, David Uminsky, James von Brecht. Multiclass Total Variation Clustering. In NIPS 2013.
[Ref3] Zhirong Yang, Jukka Corander and Erkki Oja. Low-Rank Doubly Stochastic Matrix Decomposition for Cluster Analysis. Journal of Machine Learning Research, 17(187): 1-25, 2016.

---

> ### Author Response · Authors · 2017-12-08
> **Response**
>
> Thank you for the helpful comments!
>
> Regarding the points that you have made:
>
> 1: The NAT training framework aims to match each latent representation to a unique target in the latent space. By doing this, the model learns a mapping to latent space in which the distribution of the latent representations of the dataset very closely matches the distribution of the noise-targets. This is done by jointly learning the encoder function (parameterized by a neural network) and also learning the assignment of examples to their best-fitting target in latent space.
>
> As we do not know the ideal assignment at the beginning of training, we randomly assign each example a noise-target at the beginning of training. During training, we progressively re-assign labels to different examples in the dataset so as to minimise the total distance between latent representations their corresponding targets (we call these optimal assignments). To find the optimal assignments, we can compute the distance from every latent representation to each target in the dataset and use the hungarian algorithm to find the optimal assignment of latent representation to target. However, finding the optimal assignment for an entire dataset of latent representations and noise-targets would be very expensive (indeed the hungarian algorithm has O(n^3) complexity), so instead we train using randomly selected batches from the dataset (we use a batch size of 100). For a batch of latent representations and targets, calculating the optimal assignments is feasible, and also means we can train NAT models similarly to other deep learning models (i.e. mini-batch SGD).
>
> We agree that the section discussing the NAT training framework is quite brief. We decided to remove a lot of discussion to reduce the paper down to 8 pages. We plan to include more discussion on the NAT framework in an upcoming revision.
>
> 2: The NATAC model does rely on some heuristics, which means we do not have analytic guarantees to this method. We set the latent space of our model to be the surface of a d-dimensional sphere (similarly to the work of Bojanowski and Joulin). Although this might be less expressive than an un-normalized latent space, we found that placing both the noise targets and the latent representations on the manifold is empirically much more effective for training (see Appendix C.1).
>
> 3: The value of lambda and alpha do change during training. There is some discussion in the appendix about how exactly we set these values. In these experiments, we used some trial-and-error to find fitting values. We aim to include some experiments showcasing how the clustering algorithm behaves with different values for lambda (and alpha). Expect an update to the paper including this soon.
>
>
> 4: Thank you for bringing these papers to our attention. We will certainly include these in our revision. We believe a key contribution of the paper is that our method does not need a prior number of clusters - as real-world use cases for clustering usually have no prior knowledge of the true number of clusters in the data. However it is clear that several methods of unsupervised clustering (which do require a given number of clusters) outperform our method on MNIST, which we will mention in our revision.
> Regarding 4 c) - Note that the evaluation of NATAC in our paper is different to that in the DCD paper: The NMI scores we report for the experiments are taken from the test set of 20 Newsgroups after training on the train set (using the ‘bydate’ version of the dataset). The DCD paper cites 20K training examples for it’s train set - suggesting that the clustering was performed using both the train and test set, with NMI reported on the whole dataset. If that is the case, we will report NMI values for our method trained in this way, and compare the results to those mentioned in the paper.
>
>
> 5: During training, the model successively agglomerates examples to the same centroid by deleting an example’s assigned target and instead assigning the example a copy of another target (using the delete-and-copy mechanism). At the same time, the model is also trying to optimize to the auxiliary objective, by having as little reconstruction error as possible.
> This means that, at some point, the model does not delete any more centroids during training, as agglomerating any more points would incur a huge reconstruction loss penalty. Therefore the model converges onto the number of clusters during training. We discuss convergence of the model in section 2.3 (Implementation Details).

---

### Official Review · AnonReviewer1 · 2017-11-27
**Interesting work but lack of detailed discussions and thorough quantitative results**

**Rating:** 5
**Confidence:** 4

**Review:**

This paper proposes a neural clustering model following the "Noise as Target" technique. Combining with an reconstruction objective and "delete-and-copy" trick, it is able to cluster the data points into different groups and is shown to give competitive results on different benchmarks.

It is nice that the authors tried to extend the "noise as target" to the clustering problem, and proposed the simple "delete-and-copy" technique to group different data points into clusters. Even tough a little bit ad-hoc, it seems promising based on the experiment results. However, it is unclear to me why it is necessary to have the optimal matching here and why the simple nearest target would not work. After all, the cluster membership is found based on the nearest target in the test stage.

Also, the authors should provide more detailed description regarding the scheduling of the alpha and lambda values during training, and how sensitive it is to the final clustering performance. The authors cited the no requirement of "a predefined number of clusters" as one of the contributions, but the tuning of alpha seems more concerning.

I like the authors experimented with different benchmarks, but lack of comparisons with existing deep clustering techniques is definitely a weakness. The only baseline comparison provided is the k-means clustering, but the comparisons were somewhat unfair. For all the text datasets, there were no comparisons with k-means on the features learned from the auto-encoders or clusterings learned from similar number of clusters. The comparisons for the Twitter dataset were even based on character-level with word-level. It is more convincing to show the superiority of the proposed method than existing ones on the same ground.

Some other issues regarding quantitative results:
- In Table 1, there are 152 clusters for 10-d latent space after convergence, but there are 61 clusters for 10-d latent space in Table 2 for the same MNIST dataset. Are they based on different alpha and lambda values?
- Why does NATAC perform much better than NATAC-k? Would NATAC-k need a different number of clusters than the one from NATAC? The number of centroids learned from NATAC may not be good for k-means clustering.
- It seems like the performance of AE-k is increasing with increase of dimensionality of latent space for Fashion-MNIST. Would AE-k beat NATAC with a different dimensionality of latent space and k?

---

> ### Author Response · Authors · 2017-12-08
> **Response**
>
> Thank you for the insightful comments!
>
> Regarding the need for the Hungarian algorithm in our model:
> We only use the Hungarian algorithm during the first two stages of training, i.e. during the warm-up and transition stages. Subsequently, assignment between targets and latent representations is done purely by assigning a target to its nearest latent representation.
>
> The aim early on in training is to (1) pre-train the encoder and decoder networks and (2) have the latent representations distribute (close to) uniformly across the latent space. We ensure this by using the NAT objective which makes the model minimise distances between latent representations and their targets.
>
>
> In contrast to the above, were we to assign targets to their nearest latent representations from the very beginning, we would risk the model collapsing in on itself as the encoder function would not have learned a stable mapping to latent space. We can corroborate this empirically through the ample runs we made early on whilst working on this paper.
>
>
> With regard to your comments on the tuning of the alpha and lambda values:
> To give an indication of the robustness of our method we plan to include experiments highlighting how much the lambda (and alpha) values affect training on MNIST. In short, we do not tune the value of alpha very much in our experiments (0 at the beginning, then a gradual increase to 1 after some epochs). However drastically different values, for example alpha set to 1 throughout training, do lead to poor results.
>
> With regards to the text datasets:
> We wholeheartedly agree that the baselines used in the text-based experiments are quite weak. Our intention was not to prove that our method is optimal for text clustering; rather, we wanted to show that the technique generalizes across modalities.
>
>
> Given your feedback, we plan on including the results from models of an identical architecture trained as vanilla autoencoders (AE-k) and k-means using the learned representations of the NATAC model. Additionally, we plan to train a model on the whole of  20-newsgroups and compare the NMI to clustering algorithms that require a predefined number of clusters. Stay tuned, the results should be in within a week.
>
> You rightfully point out the discrepancy between the number of clusters for the model in Table 1 and Table 2. We use the same hyperparameters for both of these experiments, however the model in Table 2 is trained on the whole of MNIST, whereas the model in Table 1 is trained on the train and validation sets only (to allow for evaluation on the test set). We plan to include some experiments which show the variability of our training method (converged number of clusters, NMI score, similarity to clusters in other models.). These are the next priority after including the updated text-dataset results
>
> The question of whether the same number of clusters would be optimal for NATAC-k is an interesting one. Indeed, seeing as many of the clusters in the NATAC models contain very few examples (the ‘dead centroids’), it would be a little unfair to compare using k-means with the same number of clusters. We are currently discussing what a more sensible baseline might be.

---

### Public Comment · (anonymous) · 2017-11-11
**Request for citation**

I believe that you should also cite “Learning Discrete Representations via Information Maximizing Self-Augmented Training” (ICML 2017) http://proceedings.mlr.press/v70/hu17b.html.
This paper is closely related to your work and is also about unsupervised clustering using deep neural networks.
As far as I know, the proposed method, IMSAT, is the current state-of-the-art method in deep clustering (November 2017). Could you compare your results against their result?

---

> ### Author Response · Authors · 2017-11-16
> **RE: Request for citation**
>
> We will certainly compare the results of the paper, along with other related clustering papers in the ICLR review, to our approach.

---

### Author Response · Authors · 2017-11-16
**Including Related Work**

Since submission, several papers have come to our attention which we would like to include and discuss:

* Learning Discrete Representations via Information Maximizing Self-Augmented Training
* Deep Continuous Clustering (ICLR 2018 submission)
* Spectralnet Spectral Clustering Using Deep Neural Networks (ICLR 2018 submission)
* Learning Latent Representations In Neural Networks For Unsupervised Clustering Through Pseudo Supervision And Graph Based Activity Regularization (ICLR 2018 submission)

Additionally, some of the above report higher NMI scores than our model (although they require a set number of clusters). We will adapt the paper respectively.

---

### Author Response · Authors · 2017-12-27
**Changes To The Paper**

We have made some changes and additions to the paper during this rebuttal/discussion period. Our main changes are to add further experiments to demonstrate the robustness of the NATAC training method, and to add more baselines to our text-based experiments. In full, we have:

* Added other clustering methods into the MNIST comparison table.
* Updated the 20news NATAC results - we have found some slightly better performing hyperparameters.
* Included NATAC-k and AE-k Results for both the 20 Newsgroups and Twitter Datasets.
* Included a Comparison table with some other clustering algorithms for 20 Newsgoups, we perform competitively although our model converges on significantly more clusters.
* Added an experiment to empirically show that the NATAC training method is fairly stable wrt final NMI and converged number of clusters. We only had the time to train multiple runs of models on 20 Newsgroups - so we were unable to officially comment on the other datasets.
* Added an experiment to show how the end performance of a model changes with increasing amounts of pre-training.
* Added an experiment to show that changes in the value of lambda (increase/decrease by a power of 10) do not greatly affect the end performance of the model.
* Altered NATAC training so it is a bit more intuitive to understand.
* Added more discussion about the NAT training framework.
* Made some small edits in the introduction and conclusion.
* Redone the Twitter dataset experiments. We used slightly different hyper-parameters which converged to fewer clusters.

---

### Decision · Program_Chairs · 2018-01-29
**ICLR 2018 Conference Acceptance Decision**

**Decision:**

Reject

**Comment:**

The paper proposes an approach to jointly learning a data clustering and latent representation.  The main selling point is that the number of clusters need not be pre-specified.  However, there are other hyperparameters and it is not clear why trading # clusters for other hyperparameters is a win.  The empirical results are not strong enough to overcome these concerns.